# An Algorithm Framework for Drug-Induced Liver Injury Prediction Based on Genetic Algorithm and Ensemble Learning

**DOI:** 10.3390/molecules27103112

**Published:** 2022-05-12

**Authors:** Bowei Yan, Xiaona Ye, Jing Wang, Junshan Han, Lianlian Wu, Song He, Kunhong Liu, Xiaochen Bo

**Affiliations:** 1Department of Bioinformatics, Institute of Health Service and Transfusion Medicine, Beijing 100850, China; boweiyan2020@gmail.com (B.Y.); hanjunshan01@163.com (J.H.); 2Department of Biotechnology, Institute of Radiation Medicine, Beijing 100850, China; 3School of Informatics, Xiamen University, Xiamen 361005, China; 24320191152535@stu.xmu.edu.cn; 4School of Medicine, Tsinghua University, Beijing 100084, China; j-w20@mails.tsinghua.edu.cn; 5Institute of Medical Engineering and Translational Medicine, Tianjin University, Tianjin 300072, China; wulianlian07@163.com

**Keywords:** DILI, genetic algorithm, ensemble learning, PCA/MCA, QSAR, molecular representation

## Abstract

In the process of drug discovery, drug-induced liver injury (DILI) is still an active research field and is one of the most common and important issues in toxicity evaluation research. It directly leads to the high wear attrition of the drug. At present, there are a variety of computer algorithms based on molecular representations to predict DILI. It is found that a single molecular representation method is insufficient to complete the task of toxicity prediction, and multiple molecular fingerprint fusion methods have been used as model input. In order to solve the problem of high dimensional and unbalanced DILI prediction data, this paper integrates existing datasets and designs a new algorithm framework, Rotation-Ensemble-GA (R-E-GA). The main idea is to find a feature subset with better predictive performance after rotating the fusion vector of high-dimensional molecular representation in the feature space. Then, an Adaboost-type ensemble learning method is integrated into R-E-GA to improve the prediction accuracy. The experimental results show that the performance of R-E-GA is better than other state-of-art algorithms including ensemble learning-based and graph neural network-based methods. Through five-fold cross-validation, the R-E-GA obtains an ACC of 0.77, an F1 score of 0.769, and an AUC of 0.842.

## 1. Introduction

Drug-induced liver injury (DILI) is a common cause of liver injury. It accounts for more than 50% of acute liver failure cases in the clinic [1]. Meanwhile, DILI is the major reason for drug failure, accounting for approximately 40% of the failed drugs in drug development. As of 2011, more than 50 types of drugs with DILI have been withdrawn from the market [2,3]. These problems cause tremendous pain and economic costs to society. To address this issue, the main toxicity assessment methods have been roughly divided into two categories, methods based on animal experiments and computational models. Compared with early animal experimental methods, the computational models not only have the advantages of time-saving, low cost and high efficiency but also are not limited by the experimental environment error [4]. Therefore, it is of vital importance to develop an efficient and accurate model for DILI prediction.

With the development of artificial intelligence, machine learning (ML) and deep learning methods have gradually shown more potential than the expert estimation methods and other statistical models in the early stage of toxicity risk assessment [5]. Among the computational models in toxicity prediction, molecular representation-based ensemble learning and deep learning methods have achieved wide application on numerous DILI datasets [6,7,8,9,10]. The molecular representation-based methods usually predict the risk of DILI and other toxicities based on quantitative structure-activity (QSAR) and molecular geometric deep learning (e.g., molecular graph representation) [11,12]. Using 21 QSAR models, Mulliner et al. predicted DILI based on a large-scale dataset consisting of 3712 compounds and achieved the area under the receiver operating characteristic curve (AUC) values between 0.71 and 0.75 [7]. He et al. obtained a prediction accuracy (ACC) of 0.783 with 1253 compounds using the ensemble learning model with eight base classifiers [8]. Wang et al. and Ai et al. obtained an ACC of 0.77 and 0.71 using the voting ensemble learning by fingerprints-based QSAR models, respectively [9,10]. The Graph neural network has been proved to be the most advanced model in many fields, such as molecular attribute prediction [13,14] and has also achieved high prediction accuracy in DILI prediction [15,16]. At the same time, ML models based on toxicogenomics have also achieved high accuracy [2]; however, their accuracy is highly dependent on experimental data, and the experimental accuracy on large datasets is not satisfactory (the best-obtained accuracy is 0.7) [17].

Among many machine learning models, the Genetic Algorithm (GA) naturally avoids some problems commonly encountered by other optimization algorithms. GA is an evolutionary algorithm, whose working principle is inspired by natural inheritance and survival of the fittest [18,19]. In the large and complex search space, GA can quickly approach the approximate global optimal solutions without getting stuck in the local optimal [20,21]. GA will evolve to the last individual as an approximate global optimal solution. As an optimization algorithm, GA has shown excellent performance in various fields, such as micro-expression recognition [22] and microarray datasets [22,23,24]. At the same time, it has strong robustness and can be combined with other algorithms for solution search optimization [25,26,27,28]. In addition to the Genetic Algorithm, the application of various optimization algorithms also reduces parameter sensitivity and maintains excellent performance [29], such as the optimization algorithm for tuning fuzzy control systems [30], and the meto-heuristic gray wolf optimizer (GWO) algorithm to train neural networks [31]. In the drug discovery problem, the solution to the problem can be encoded as an individual so that the better-performing individuals can be screened out through population evolution. The solutions with better performance can be selected globally. For example, taking the feature combination as the solution to the problem, the feature combination with excellent performance can be screened out through GA iterative evolution. Because of its string-structure individual, it can provide the linear representation for different fields and has been successfully applied to tackle a set of optimization problems in various research fields, such as feature selection [32], ensemble learning [33], and unsupervised learning [34]. GA has also been applied to solve complex medical problems, including disease screening [35], disease diagnosis [36], prognosis [37], and health care management [38]. In drug discovery, GA has been used in the evaluation of adverse drug reactions [39], drug design, and lead optimization [40,41,42].

In this study, we proposed an algorithm framework, Rotation-Ensemble-GA (R-E-GA), based on GA, Rotation Forest and Ensemble Learning to solve the problem of DILI prediction. The Rotation in R-E-GA improves the rotation operation in Rotation Forest. The rotation operation refers to the rotation transformation of the feature space to search for a better data space. Previous studies have applied Rotation Forest for scheme design and achieved good results [43]. For DILI data in this study, we first collected, sorted, and cleaned a large number of compounds containing DILI labels from different literature sources, and combined them with the classical DILI research dataset to ensure overall data quality. Then, we constructed our data by calculating and splicing different types of molecular fingerprints (ECFP, MACCS, Rdkit2D, etc.), which fully considered the molecular spatial information of compounds. These fingerprints contain circular fingerprints, sub-structural key fingerprints and pharmacophore descriptors, including both continuous variables (Rdkit2D) and discrete variables (other fingerprints). Traditional ML methods find it difficult to extract toxicity features from such high-dimensional and complex chemical spaces. Compared with traditional ML and ensemble models, rotation GA first uses PCA to flip the feature space and extract toxic fingerprints and features. Next, genetic algorithm and ensemble learning are combined to further improve the accuracy and generalization ability of the model. Then, we compared the prediction performance of the R-EGA framework, graph neural network, and classical machine learning algorithms on this dataset. Finally, we applied the computational features (One-hot encoder, Mold2, PaDEL), and label dataset proposed by Xu et al. [6]. for model training and external validation. The results show that the performance of R-E-GA is superior to their algorithm in molecular descriptors. According to the results, our algorithm achieves the highest DILI prediction accuracy.

## 2. Materials and Methods

### 2.1. Datasets

We first collected a large number of compounds labeled with human DILI data from public databases, as shown in Table 1. We processed the collected data in the following steps to finally form a DILI signature dataset containing 2931 compounds, including 1498 positive and 1433 negative:(1)The Python 3.8 Rdkit tool [44] was used to match Canonical SMILES from the literature through SMILES, and the corresponding Pubchem Compound ID (CID). We combined the DILI-tagged data from different authors, and removed duplicate data as well as metals and compounds containing rare elements.(2)We binarized the labels of different datasets to obtain binary labels. The rules of the label are shown in Table 1. We adopted cautious binarization rules and took compounds with high reliability DILI classes. First, the data came from trusted sources, such as scientific literature, medical monographs clinical data and data approved by the FDA. Second, we set labels to “1” for the compounds with definite DILI from the original source, and “0” for the compounds without DILI from the original source. This is reflected in the processing of Greene, DILIrank, Livertox and LTKB data, where “HH” and “Most concern” represent “Evidence of human hepatotoxicity” and DILI-positive, respectively. Meanwhile, “NE” and “no concern” indicate “no evidence of hepatotoxicity in any species” and DILI-negative [45,46,47,48,49,50]. The “Category A” and “Category B” from the LiverTox are the classes of compounds that have been “frequently reported” and “reported” to cause DILI; “Category E” means “no evidence that the drug has caused liver injury” [47,51]. We found that Shuaibing et al. and Mulliner et al. had the same strict binarization rules we adopted, so we considered the data of these authors also to be credible [7,8]. It was found that Xu et al.’s binding data also came from highly trusted data sources, including NCTR, Greene et al., and Xu et al., which removed inconsistent compound’s DILI label from the dataset, and we considered their data equally reliable [6]. Finally, to expand the dataset, we took a small portion of compounds from Greene’s “WE” compound’s DILI classes which represented “Weak evidence of (<10 case reports) human hepatotoxicity”, and they were also considered as DILI compounds in the literature [47,50].(3)Since the toxicity labels of compounds in different datasets may be inconsistent, we first retained the more reliable data sources from the three large databases, DILIrank [46], LiverTox [47], and LTKB [48], and then removed the corresponding compounds from other datasets [8].(4)We voted on the remaining data to determine the label of the compound. The voting rules were as follows: if the label of a compound is consistent in all datasets or consistent in 80% of the datasets, we take the label as the toxicity label of the compound; otherwise, we delete the compound.

**Table 1 molecules-27-03112-t001:** Datasets of DILI and binarization rules of labels.

ID	Source	Type of Data	No. of Compound (Positive/Negative)	DILI Categories
1	Greene et al., 2010 [45]	Literature reviews and medical monographs	487 (331/156)	HH, WE represented positives and NE represented negatives
2	Xu et al., 2015 [6]	Medical monographs and FDA-approved drug labeling	475 (236/239)	Authors definition
3	Mulliner et al., 2017 [7]	Clinical data and drug labeling	1370 (932/438)	Authors definition
4	Shuaibing et al., 2019 [8]	Drug labeling and comprehensive data	1458 (761/697)	Authors definition
5	DILIrank [46]	Drug labeling and clinical data	504 (192/312)	Most concern as 1; no concern as 0
6	Livertox [47]	Scientific literature and public database	343 (119/224)	Categories A and B were combined into positives, and Category E was considered as negatives
7	LTKB [48]	FDA-approved drug labeling	195 (113/82)	Most concern as 1;no concern as 0

After the above steps, the DILI dataset was expanded and the accuracy of the data was ensured, and the prediction accuracy in the baseline was basically consistent with that reported in other literature. The dataset used in this paper is included in the Appendix A.

### 2.2. Molecular Representations

In order to obtain a better molecular representation for models, we first calculated eight molecular fingerprints (descriptors) of compounds using Python Rdkit tools, including extended connectivity fingerprints, structural keys fingerprints, and pharmacophore descriptors, as shown in Appendix A. Then we used the traditional ML algorithms to predict DILI for the pre-experiment with different molecular fingerprints (descriptors), the results of ACC for prediction performance are shown in Figure 1 (more details of the results are shown in Appendix A). These algorithms include Random Forest (RF) [52], Support Vector Machine Classifier (SVC) [53], Extreme Gradient Boosting (Xgboost) [54], Gradient Boosting Decision Tree (GBDT) [55], Adaptive Boosting (Adaboost) [56], Logistic Regression (LR) [57] and Decision Tree (DT) [58]. After our pre-experiment, we selected three types of molecular fingerprints and spliced them as the input of the models. They are Rdkit2D, PubChem, MACCS, and ECFP2 fingerprints. Rdkit2D, ECFP, and MACCS have been shown to achieve comparative predictions with graph neural network-based methods in some toxicity prediction tasks [15]. Pubchem fingerprints show similar prediction accuracy to the ECFP and MACCS fingerprints in our study, so we spliced these four fingerprints together as the model input. For toxicity prediction, the multi-fingerprint method has a higher prediction accuracy than the single-fingerprint method [9,10,59]. For the graph embedding model, we used the information of atoms and bonds of compounds to construct molecular graphs, which is consistent with the literature [13,60] and constructed by Python 3.8 DGL-LifeSci-0.7.0 [61].

### 2.3. GA Algorithm

GA is a stochastic global search optimization method. It simulates the phenomena of crossover, mutation, and selection that occur in natural selection and genetics. Starting from a random initial population, through random selection, crossover, and mutation operations, a group of individuals more suitable for the environment is generated.

The terms of GA are defined as follows:Individual: A solution to a problem, and a unit of evolution.Bit: A code that constitutes the solution to the problem.Fitness: The degree of individual adaptation to the environment.

The workflows of GA are as follows:Encoding: The mapping from the solution of the problem to the individual.Decoding: The conversion of the individual to the problem solution.Initialization: Set the maximum evolutionary T, population size M, crossover probability $P_{C}$, mutation probability $P_{M}$, and randomly generate M individuals as the initial population $P_{0}$.Fitness: The fitness function indicates the pros and cons of the individual or solution. Genetic Operator: Three types: selection operator, crossover operator, and mutation operator. Each population $P_{t}$ is manipulated by the genetic operators to obtain the next generation $P_{t+1}$.Termination: When the evolution generation reaches the maximum *T*, the evolution is terminated.

### 2.4. Framework of Rotation-Ensemble-GA Algorithm

R-E-GA is a GA-based ensemble learning framework with the aid of feature rotation. It follows the workflow of GA, including population initialization, crossover and mutation operation, fitness value evaluation, elite retention, and loop iteration. First, it initializes the population and generates M individuals randomly as the primary population according to the encoding method. Then the population is processed by the crossover and mutation operators to generate offspring, obtaining a total of 2×M individuals from both parent and offspring populations. The fitness of the 2×M individuals is calculated through the fitness function, and then M individuals with higher fitness are selected and retained to form a new population. Before reaching the maximum evolutionary generation number T, the new population is operated by the crossover and mutation step to continue the loop. When T is reached, the loop is terminated and the final generation is the high-quality solutions obtained by our algorithm.

Among them, the part of feature subspace rotation and ensemble learning is applied in the fitness function, i.e., the *Evaluation Fitness Value Part* in Figure 2. The unit for calculating fitness value is an individual, which consists of K feature subsets. The K feature subsets are all rotated by principal component analysis (PCA) or multiple correspondence analysis (MCA) to form K new feature subsets, which are applied to train *K* weak classifiers used for ensemble learning. The ensemble learning method adopted by R-E-GA is a boosting method similar to Adaboost. First, it trains a weak classifier with the initial weights of the training samples. Next, it updates the weights according to the error rate of the weak classifier to train the next weak classifier based on the training set. This process continues until K weak classifiers are generated sequentially. Finally, these *K* weak classifiers are integrated through an ensemble strategy to obtain the final ensemble represented by the related individual.

The rotation step in the R-E-GA is inspired by the Rotation Forest algorithm, which mainly refers to dividing all features into K equal-sized subsets, and then using PCA to rotate the feature subset. It is well known that the variance of different classes becomes larger in the rotated feature space, making the classification task much easier. However, compared with the rotation effect of the feature space containing all features, the feature subspace formed by a part of appropriate features often leads to better discriminative capability, improving the quality of the data and the performance of the model. GA provides a great global search for proper K feature subsets, based on which the solution of the R-E-GA is defined as the division method of K feature subsets. An individual represents an ensemble classifier containing K weak classifiers, fused by the Adaboost strategy.

In R-E-GA, each ensemble is represented as an individual for evolution. New solutions are found through the crossover and mutation operations. The individuals with better performance are selected and retained. In this way, the solution in the candidate set is in a process of gradual optimization. The search process finds a better feature subset through continuous evolution, evolving better ensembles. When the evolution ends, the optimal individual is regarded as the approximate global optimal solution. Because of the randomness of each operation in GA, it is guaranteed that the algorithm tends to generate an optimal solution close to the global optimum.

### 2.5. Details about R-E-GA

#### 2.5.1. Initialization

R-E-GA defines the solution to the problem as K feature subsets suitable for rotation. Therefore, each bit of an individual indicates a feature index. Each feature subset consists of $S_{fs}$ feature indices, so each individual consists of K feature index sets of size $S_{fs}$, which may contain duplicate feature indexes. The following Algorithm 1 is the pseudo-code of initialization.
**Algorithm 1** Initialization**Input:** M: the amount of individuals in a population.K: the amount of feature subsets in an individual.$S_{fs}$: the size of feature index in a feature subset.$S_{f}$: feature size.**Output:** $P_{0}$: the first population of R-E-GA1: for m=0……M do2:   m=m+13:   reset feature_subset to []4:   while k=0……K do5:   k=k+16:   add the feature_index randomly generated in $S_{f}$ range to feature_subset7:   add feature_subset to $P_{0}$

#### 2.5.2. Crossover and Mutation

The crossover operator randomly exchanges the effective information between individuals to reorganize the individuals, which is beneficial to search for better solutions in the solution space. A specific crossover operation is described in Figure 3a. Two individuals in the population are randomly selected first. Then a position is picked up as the intersection point, and the two parts of the two individuals are exchanged by the intersection point to form two offspring. The iteration continues until all individuals have undergone the crossover operation to form the initial offspring.

The mutation operator aims to randomly change several codes to introduce new feature combinations. Modifying a small proportion of several feature indexes on the existing feature subsets allows for the exploitation of new solution space. Therefore, the mutation operator is used to introduce new information, explore new coding possibilities, and generate new solutions. The mutation operator has a random search direction, which is conducive to jumping out of the local optimal solution. Figure 4 is an example of a mutation of a subset of features in an individual.

The pseudo-code of the whole process of Crossover and Mutation Algorithm 2 is as follows.
**Algorithm 2** Crossover and Mutation**Input**: $P_{i}$: *i*th population. 
   $P_{M}$: the possibility of mutation. 
   M: the amount of individuals in a population. 
   K: the amount of feature subsets in an individual. 
   $S_{f}$: feature size. 
**Output**: C: generated newly population after crossover and mutation operation. 
   1:  C←  [] 
   2:  for i=0…M/2 do 
   3:    select two individuals $I_{i}$ and $I_{j}$ from Pi randomly 
   4:    select crossover point CP in 0 to K randomly 
   5:      CP divides $I_{i}$ and $I_{j}$ into left and right parts, i.e., $I_{i\_left}$,$\;I_{i\_right}$, $I_{j\_left}$, $\;I_{j\_right}$, respectively 
   6:     $C_{i}$ $=I_{i\_left}+I_{j\_right}$
   7:     $C_{j}=I_{j\_left}+I_{i\_right}$
   8:     add C_i and C_j to C. 
   9:  for individual I in Children do 
   10:    select a number of mutation bits MBS with the probability P_M 
   11:    for b in MBS do 
   12:     I [b] = random number in S_f

#### 2.5.3. Fitness Function 

Figure 3b shows the fitness evaluation process of an individual. The fitness value is a very important step in GA, equivalent to the objective function of the problem. The specific fitness function is described as follows.

First, each feature subset is divided into continuous features and binary features, and then the rotation operation is performed by applying the principal component analysis (PCA) to the continuous feature and the multiple correspondence analysis (MCA) to the binary feature. Then, the two parts of the features are merged to accomplish the rotation operation of a feature subset. After the rotation, each feature subset needs to train its weak classifier subsequently. The training of the first classifier uses all samples with the weight defined in Equation (1). The sample weights are adjusted according to the results of the first classifier in the training set by Equation (2), and the weight of the wrong samples is increased according to Equation (4). The second classifier is trained based on the re-assigned sample weights and the second feature subset, and so on. In this way, each classifier is trained based on the training of the previous classifier, which is based on the Adaboost scheme. Finally, all classifiers make predictions about the validation set, and the weight of each classifier’s vote is assigned by Equation (3) according to the error rate of each classifier on the training samples. The voting rules are shown as Equation (5). The F1-score calculated by Equation (8) after weighted voting is used as the fitness value of the individual.
(1)$$w^{1}=\left[w_{1},\ldots ,w_{N}\right],\;w_{j}^{1}=\frac{1}{N}.\;\sum_{j=1}^{N}w_{j}^{1}=1$$
(2)$$\epsilon_{k}=\sum_{j=1}^{N}w_{j}^{k}l_{k}^{j},\;(l_{k}^{j}=1\;\mathrm{if}\;D_{k}\;\mathrm{misclassifies}\;t_{j}\;\mathrm{and}\;l_{k}^{j}=0\;\mathrm{otherwise}.)$$
(3)$$\beta_{k}=\frac{\epsilon_{k}}{1-\epsilon_{k}},\mathrm{where}\;\epsilon_{k}\in \left(0,0.5\right)$$
(4)$$w_{j}^{k+1}=\frac{w_{j}^{k}\beta^{\left(1-l_{k}^{j}\right)}}{\sum_{i=1}^{N}w_{i}^{k}\beta^{\left(1-l_{k}^{i}\right)}},\;j=1,\ldots ,N.$$
(5)$$\mu_{t}\left(x\right)=\underset{D_{k}\left(x\right)=w_{t}}{\sum }\ln\left(\frac{1}{\beta_{k}}\right)$$
(6)$$precision=\frac{TP}{TP+FP}$$
(7)$$recall=\frac{TP}{TP+FN}$$
(8)$$\left\{\left.\begin{array}{cc} TP:\;The\;number\;of\;samples\;that\;are\;actually\;positive\;and\;predicted\;to\;be\;positive\\ FP:\;The\;number\;of\;samples\;that\;are\;actually\;negative\;and\;predicted\;to\;be\;positive & FN:\;The\;number\;of\;samples\;that\;are\;actually\;negative\;and\;predicted\;to\;be\;negative\\ TN:\;The\;number\;of\;samples\;that\;are\;actually\;positive\;and\;predicted\;to\;be\;negative \end{array}\right\}\right.$$

The pseudo-code of the fitness function Algorithm 3 is as follows.

**Algorithm 3** Fitness Function**Input:**N: The samples amount of training set. K: The amount of feature subsets in an individual. The number of weak classifiers to train.I: An individual.T: Training Set.**Output**: V: Fitness value of I.
1:    For k=1,…,K do
2:    Divide features into Binary Feature FB and Continuous Feature FC.
3:    Rotation: Apply PCA to FC and apply MCA to FB and then merge the two parts.
4:    Initialize the weights $w^{1}=\left[w_{1},\ldots ,w_{N}\right],\;w_{j}^{1}=\frac{1}{N}.\sum_{j=1}^{N}w_{j}^{1}=1.$

5:    For k=1,…,K do
6:    Take a sample $S_{k}$ from T using distribution $w^{k}$.
7:    Train a classifier $D_{k}$ using $S_{k}$ as the training set.
8:    Calculate the weighted ensemble error at step k by $\epsilon_{k}=\overset{N}{\underset{j=1}{\sum }}w_{j}^{k}l_{k}^{j},$ ($l_{k}^{j}=1$ if $D_{k}$ misclassifies $t_{j}$ and $l_{k}^{j}=0$ otherwise.) 
9:    If $\epsilon_{k}=0\;or\;\epsilon_{k}\geq 0.5$, ignore $D_{k}$, reinitialize the weights $w_{j}^{k}$ to $\frac{1}{N}$ and continue.
10:     Else, calculate $\beta_{k}=\frac{\epsilon_{k}}{1-\epsilon_{k}},$ where $\epsilon_{k}\in \left(0,0.5\right)$.
11:     Update the $k^{th}$ part weights in I by $w_{j}^{k+1}=\frac{w_{j}^{k}\beta^{\left(1-l_{k}^{j}\right)}}{\sum_{i=1}^{N}w_{i}^{k}\beta^{\left(1-l_{k}^{i}\right)}},\;j=1,\ldots ,N.$
12:    Calculate the support for each class $w_{t}$ in *Validation Set* by $\mu_{t}\left(x\right)=\underset{D_{k}\left(x\right)=w_{t}}{\sum }\ln\left(\frac{1}{\beta_{k}}\right).$
13:     The class with the maximum support is chosen as the label for x. 
14:     V is calculated by $F1-score=\frac{2\times precision\times recall}{precision+recall}$n validation data.

### 2.6. Experiment Settings

The dataset is randomly divided into the training set, the validation set, and the test set in a ratio of 2:1:1. The average result of the five-fold cross-validation is used as the final result of the algorithm. The preprocessing step only includes the normalization in Equation (9). The final dataset contains 2931 samples with 3296 features.
(9)$$x^{\prime }=\frac{x-\min\left(x\right)}{\max\left(x\right)-\min\left(x\right)}$$

The algorithm settings in the experiments are as follows: Among them, Iteration Size, Population Number and Possibility of Mutation are the parameters of the optimization algorithm, while Base Classifier, Ensemble Size and Feature Subset Size are not. They are all user-selectable.


Base Classifier: $RandomForestClassifier\left(n\_estimators=50\right)$Iteration Number T: 50Population Size M: 50Possibility of Mutation: 0.1Ensemble Size K: 10Feature Subset Size: 300


### 2.7. Details of the Comparison Algorithm

#### 2.7.1. Voting Ensemble

The Ensemble vote classifier combines similar or conceptually different machine learning classifiers and tries to obtain better predictive performance than the individual classifier alone. Previous articles often used soft voting ensemble models to predict DILI [8,9]. To compare the prediction performance of R-E-GA, we constructed the predictive performance of Voting Ensemble using five base classifiers in the Python 3.8 scikit-learn package [62], including RF, Xgboost, GBDT, SVC, and Adaboost. Each base classifier has the same weight in our Voting Ensemble models. We did not collect all the base classifiers mentioned in the literature, and for the prediction accuracy of the model, the number of base classifiers is not better for the performance of prediction [63]. We believe that the ensemble model based on an excessive multi-base classifier does not have a extensive generalization ability and simplicity principle. Therefore, we choose a certain base classifier to stabilize the prediction performances of the model.

#### 2.7.2. Graph Embedding Neural Networks

The graph embedding neural networks achieve a good prediction accuracy in compound attribute prediction [15]. In this study, we used the Python 3.8 DGL-LifeSci-0.7.0 package to predict DILI using four graph-based neural network algorithms, i.e., GCN (Graph Convolutional Network), AttentiveFP, GIN_AttrMasking and GIN_ContextPred [60]. AttentiveFP is a graph neural network architecture using a graph attention mechanism to learn from relevant drug discovery datasets [13]. GIN_AttrMasking and GIN_ContextPred pre-trained Graph Isomorphism Network (GIN) with Attribute Msking and Context prediction, respectively. The hyper-parameters were set as follows: hidden dims = 512, train epoch = 150, learn rate = 0.001, batch size = 128.

## 3. Results and Discussion

### 3.1. Comparison of the Results of Each Algorithm

First of all, Table 2 shows the comparison of different prediction algorithms in the data proposed by this paper. The algorithms include Random Forest, SVC, Xgboost, Voting Ensemble, AttentiveFP, GCN, GIN_AttrMasking and GIN_ContextPred.

For our datasets of 2931 compounds, it can be seen that compared with the current commonly used drug discovery algorithms, R-E-GA achieved the best results in various evaluation metrics, and each evaluation indicator was improved. Compared with the traditional machine learning method, the Voting ensemble method has a higher prediction accuracy than the basic classifier, which is consistent with the literature [8,9,10]. The graph-based neural network method shows strong competitiveness compared to the traditional fingerprint-based method and outperforms most fingerprint-based methods, reflecting the superiority of the graph neural network for molecular representation, such as GIN_ContextPred in DILI prediction. Compared with the pre-experiment, the ACC results of RF and SVC are improved, indicating that our multi-fingerprint strategy has a certain effect on DILI prediction. However, both F1-Score and AUC show a declining trend, which fully illustrates the bottleneck of the traditional model in extracting high-dimensional features and fails to predict DILI well. However, R-E-GA solved it by finding a subset of features after rotation in the feature space and achieved a better prediction performance.

### 3.2. External Validation

In this section, three computational features, i.e., One-hot encoded [6,64], Mold2 [65] descriptor, and PaDEL [66] descriptor were used for model training on the Combined validation dataset from Xu et al. [6]. The label information of Xu et al. [6] was shown in Table 3. The Combined dataset was compiled by Xu et al. with four original datasets sources, that is, NCTR [6,67], Liew [68], Greene [45], and Xu [69]. Moreover, the algorithms that Xu et al. used are the original UGRNN [64] and DNN algorithms [6]. Their main experimental results comparing three computational features were compared on the Combined dataset, and we performed the calculations under exactly the same dataset settings. The results of experiments are shown in Table 4 and Table 5.

We first compared the computational models of Xu et al. [6] based on molecular descriptors on the Combined dataset. In the original article, Xu et al. calculate two molecular Mold2 (777) [65] and PaDEL (1444) [66] for model training, respectively. We also calculated Mold2 and PaDEL descriptors as model inputs and the results to compare algorithms were shown in Table 4. We adopted the same 10-fold cross-validation and evaluation index as ACC, AUC, SEN, and SPE for modeling in the Combined dataset. The prediction results of R-E-GA on the Combined dataset with the Mold2 descriptor were an ACC of 0.852, SEN of 0.855, SPE of 0.850 on the 10-fold test dataset, and the prediction results were an ACC of 0.851, AUC of 0.949, SEN of 0.794, SPE of 0.898 on the external dataset, which are about an average 0.02 index higher than Xu’s model. For PaDEL descriptors used in this study, we also outperformed Xu’s model in the 10-fold test and external validation set as shown in Table 4. Thus, we find that R-E-GA has better performance in the descriptor-based model.

At the same time, we compare SMILES‘ one-dimensional linear representation of the one-hot coding model of Xu et al. [6,69]. In Xu et al.’s model, atom types are encoded as C = (1,0,0), N = (0,1,0), O = (0,0,1) , and bond types are similarly encoded [6,64]. The same molecular encoding method was adopted in our model, and the performance of comparison to R-E-GA on the Combined dataset was shown in Table 5. We find that the performance of this model is lower than that of Xu’s model when using SMILES ′ one-hot encoded vector for DILI prediction. We thought the One-hot Encoder method can be regarded as the representation extracted from the one-dimensional linear representation of molecules’ SMILES [6,12]. The model R-E-GA cannot extract toxicity fingerprints, such as physicochemical properties and pharmacophores from linear representations based on SMILES for DILI prediction. Meanwhile, RNN and deep learning show excellent accuracy in processing natural language or sequence data [6,64]. In addition, on the same dataset, R-E-GA based on a single descriptor Mold2 and deep learning based on a one-hot encoded model was basically consistent with index AUC (0.949 to 0.955).

We can conclude that R-E-GA performs best in descriptors of Modl2 and PaDEL. Each metric can achieve the best performance in each dataset. However, in the One-hot encoded vector for molecules, the performance is more moderate, and the results of R-E-GA are worse than Xu et al’s. The result was in line with our expectations. R-E-GA is designed as a DILI prediction model for the multiple molecular fingerprint (descriptor) fusion method, and it has obvious computational advantages for high-dimensional fusion representations data. Therefore, R-E-GA obtains better performance than Xu et al. when using Mold2 and PaDEL descriptors for model training. The above results show that R-E-GA has obvious advantages in the processing of the DILI prediction model based on molecular 2D fingerprints (descriptors), but it has limitations in the processing of linear representation. and we already proved the complexity of multiple molecular representations fusion is conducive to reflecting the superiority of R-E-GA.

### 3.3. Evolutionary Curve

According to the GA framework, the population is continuously optimized in the evolutionary process. The evolution curve depicts the change in the outcome of the best-performing individuals in each generation in the evolutionary process. Figure 5 shows the evolution curve of each fold data in the algorithm from a five-fold crossover experiment.

The fitness value in the algorithm was calculated on the validation set. Because the top half with better performance is retained each time a new population is generated, the performance of the algorithm on the validation set is monotonically increasing, as shown in Figure 5a–e. Although the performance of the test set may not necessarily improve in a short time, the performance of the test set is improved in terms of the entire evolution process. As shown in Figure 5a–d, the test set has both rising and falling trends, but the final results are all improved compared to the initial results.

### 3.4. Ablation Experiment

The feature data include features from several parts of ECFP2, MACCS, Rdkit2D, and Pubchem. In order to observe the contribution of the features of each part to the results, we conducted ablation experiments. The final experimental results are shown in Figure 6.

From Figure 6, F1 and ACC scores are relatively close. First, all three metrics achieved the best results when all the features were used. The effect of missing any part will be reduced, so in the end, all the parts of the feature are needed. Secondly, it can be seen that for the F1 score and ACC indicators, the lack of some features of MACCS has the lowest effect. For AUC, the lack of features in the Rdkit2D part is the least effective. This can indicate that the contribution of MACCS and Rdkit2D to the final effect of the experiment is slightly larger than that of ECFP2 and Pubchem.

In addition to the comparative experiment of ablating one type of fingerprint, ablating two types of fingerprints and three types of fingerprints was also performed. Figure 7 shows the experimental results using only two fingerprints. The blue bar of the ACC is close to the orange bar of the F1-score. It can be seen that the F1-score results of the two feature combinations are in the range of 0.75–0.77, and the AUC results are in the range of 0.82–0.84. The difference between the highest and the lowest of various feature combinations is two percentage points. The best performing feature combination is the fingerprint feature combination of ECFP2 and Rdkit2D, and the worst is the fingerprint feature combination of MACCS and Pubchem, which both belong to substructure fingerprints. This means that mutual information from fingerprints of the same type is minimal in our study. Figure 8 shows the result that only one type of fingerprint is kept. The F1-score result of one fingerprint is in the range of 0.74–0.76, and the AUC result is in the range of 0.81–0.84. The experimental results of the four fingerprints are not very different, and the Pubchem fingerprint feature is the best. From the comparison between various ablation experiments, it can be seen that the experimental effect of the combination of three fingerprints is one percentage point better than the experimental effect of the combination of two fingerprints. Additionally, the experimental effect of the combination of two fingerprints is better than that of only one fingerprint. However, all ablation experiments are less effective than all fingerprints. So, we finally decided to use four fingerprint features.

These results indicate that molecular fingerprints based on substructure and pharmacophores can better represent the toxic fingerprints for DILI prediction. Meanwhile, pharmacophore-based fingerprints named Pharmcoprint show that pharmacophore fingerprints are superior to other molecular fingerprints for protein targets prediction [70]. In the pre-experiment, we also found that the fingerprint based on pharmacophores had the highest accuracy in the single fingerprint experiment, but the prediction accuracy was higher when it was combined with other fingerprints by using R-E-GA. Furthermore, toxicity prediction is better if the characteristic codes of pharmacophores are taken into account, especially for models based on molecular graphs and linear molecular representations. The combination of fingerprints and pharmacophore descriptors is not new, but the R-E-GA framework proposed by us for the first time can better complete the extraction and classification calculation of the toxicity of fingerprint combination, and successfully apply it to DILI prediction.

### 3.5. The Proportion of Import Features

When the R-E-GA framework reaches the last generation, all individuals at this time are excellent solutions to the problem. The features in *K* feature subsets of all individuals in the last generation are counted. In this way, the features that contribute more to the classification effect can be screened out. The features that appear with higher frequencies are considered to be more important. We filter out the top 500 features of feature importance by this rule and count the proportion of them belonging to various types of fingerprints. As shown in Figure 9, it can be seen that the ECFP2 has the largest number of fingerprints in the top 500, followed by the Pubchem class, while the MACCS and Rdkit2D are few. However, considering that the number of features of various fingerprints originally input into the algorithm is inconsistent, it cannot be analyzed only in terms of quantity. ECFP2, MACCS, Rdkit2D, and Pubchem have 2048, 167, 200, and 881 features of various fingerprints, respectively. The ratios obtained by dividing the number of features of various fingerprints in the top 500 by the number of their original inputs are compared in Figure 10. The proportion of the four types of fingerprints is not very different in the range of 0.14–0.17. Among them, MACCS and Pubchem account for a higher proportion, while ECFP2 and Rdkit2D account for a lower proportion. This shows that the effective fingerprints in the MACCS and Pubchem features are higher than the others.

Although R-E-GA obtained better performance of DILI prediction than other machine learning algorithms in this experiment, our understanding of molecular representation is still limited by traditional molecular representations. Compared with deep learning, we find that it is hard for R-E-GA to extract toxicity fingerprints from one-dimensional linear representations, which is the same problem in current traditional machine learning. This is the impetus for our continued research on new molecular representations and toxicity prediction methods. In addition, we believe that it is difficult to greatly improve the DILI prediction accuracy by relying only on the molecular representation of compounds. We are prepared to combine molecular representations and multi-omics data to develop a multi-dimensional data fusion toxicity prediction machine learning algorithm next. The multi-omics contains transcriptional expression profiles, metabolites and target data of compounds, etc. This is a challenge for R-E-GA, but in the process of multi-dimensional data fusion, the model performance can be further improved through information complementarity, and it also provides us with the direction of model improvement.

## 4. Conclusions

This paper proposes a new method to generate and process DILI data in the process of drug discovery. Based on the data, a classification machine learning algorithm, R-E-GA, is proposed. This algorithm is based on the GA and combines the rotation operation in the Rotating Forest and ensemble learning. It designs the individual in the R-E-GA as an ensemble learning classifier containing k feature subsets. After the features are extracted from the k feature subsets, the PCA and MCA operations are performed on continuous features and binary features, respectively, to complete the rotation and obtain a feature space with a better classification effect. Then, *K* weak classifiers are trained from *K* feature subsets, and specific ensemble rules are designed for training and prediction according to the Adaboost-type ensemble method. Through our experiments, it was found that predicting DILI on a large number of datasets can achieve better experimental accuracy and generalization ability, which is consistent with the literature [8]. At the same time, the neural network model based on the molecular graph is indeed very competitive. We found that the use of multi-molecular fingerprints can better characterize compounds compared to single-molecule fingerprints, which indicated the insufficiency of existing characterization methods. Therefore, we expect to be able to characterize compounds through better molecular representation methods in the future. Finally, the solution searched by GA is considered to be the approximate global optimal solution obtained by the algorithm. In the external validation experiment, R-E-GA obtained better predictive performance than Xu et al.‘s model on the model 2 and PaDEL molecular descriptors. Experimental results showed that the R-E-GA algorithm outperforms other algorithms.

## Figures and Tables

**Figure 1 molecules-27-03112-f001:**
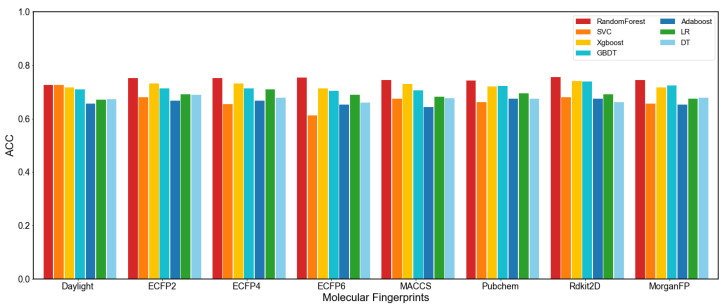
The performance of ACC in pre-experiment with different Fingerprints.

**Figure 2 molecules-27-03112-f002:**
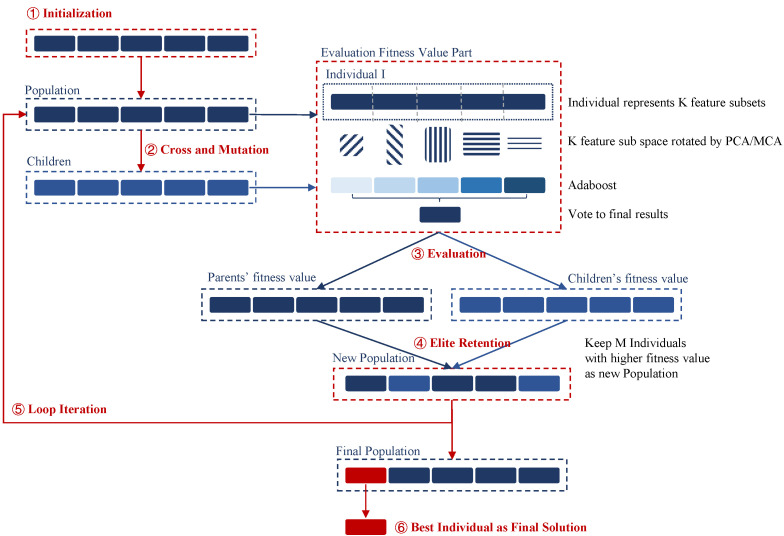
The overall framework of R-E-GA. The steps in red fonts ①–⑥ indicate the sequence flow of our algorithm, the blue fonts represent the objects and elements operated by the algorithm, and the black fonts represent some descriptions of the operation.

**Figure 3 molecules-27-03112-f003:**
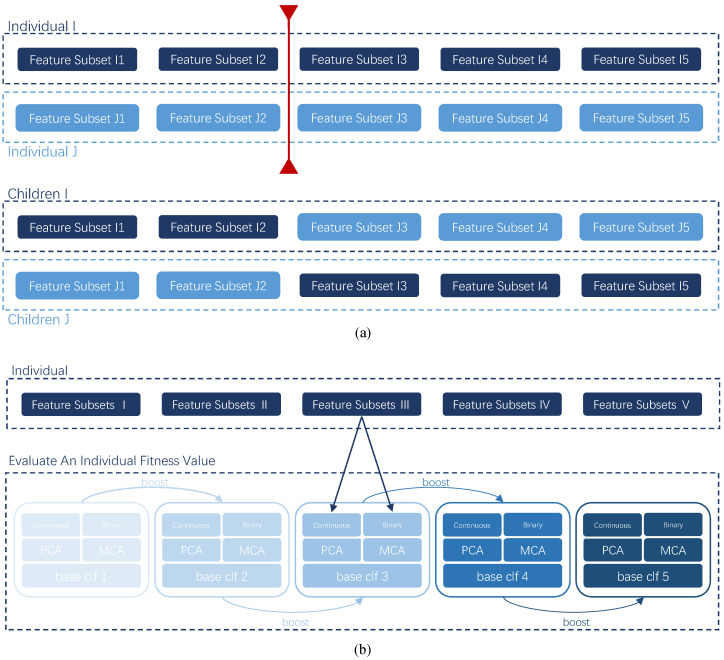
Genetic Operator: (**a**). Schematic diagram of the cross operation. Two individuals generate offspring by exchanging the parts before and after the Crossover Point. (**b**). Specific calculation flow chart of the fitness function.

**Figure 4 molecules-27-03112-f004:**
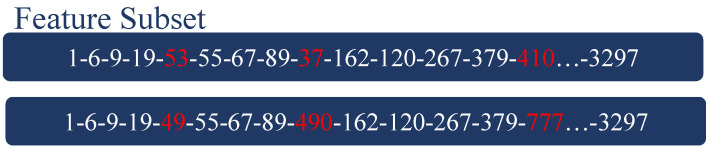
An instance of mutation in a feature subset in an individual. It randomly selects a number of codes and mutates into a certain number randomly. The limited range of this number is the number of features, which is the same as the range of individual codes.

**Figure 5 molecules-27-03112-f005:**
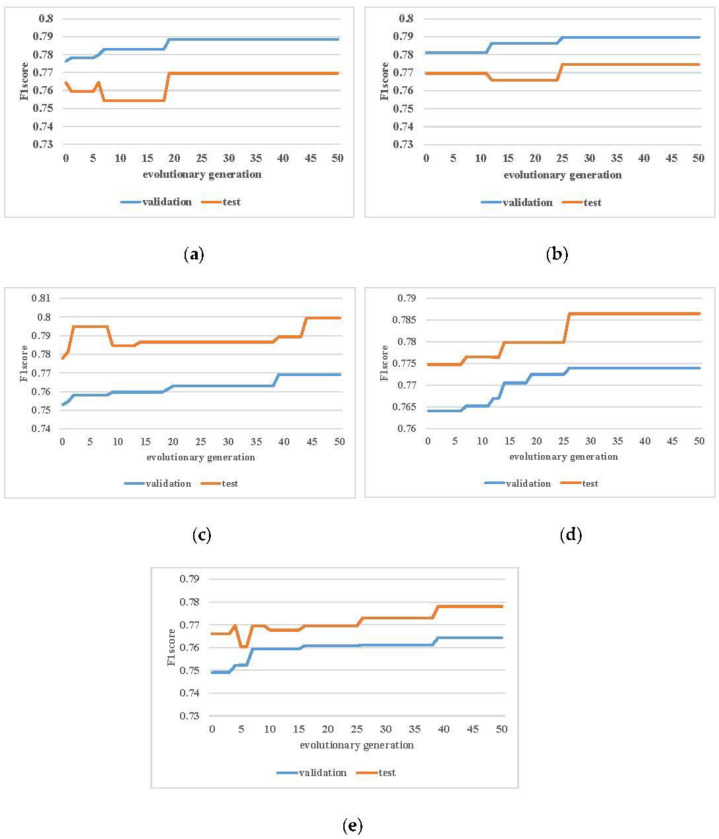
The evolution curve of each fold data in five-fold cross-validation. The blue curve represents the results of the validation dataset, and the orange curve represents the test dataset. The abscissa represents the current evolutionary generation, and the ordinate represents the F1-score of the current optimal individual. Among them, (**a**–**e**) are the evolution curves of each fold data in a five-fold cross-validation experiment.

**Figure 6 molecules-27-03112-f006:**
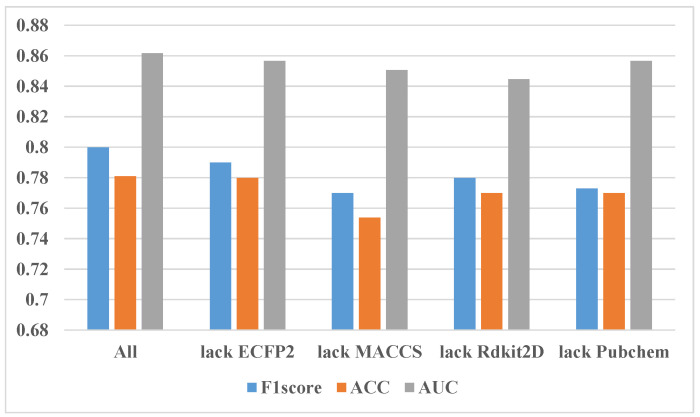
Results of ablation experiments. “All” means using all the features, “lack ECFP2” means removing the ECFP2 part of the features from all the features, and so on for other labels.

**Figure 7 molecules-27-03112-f007:**
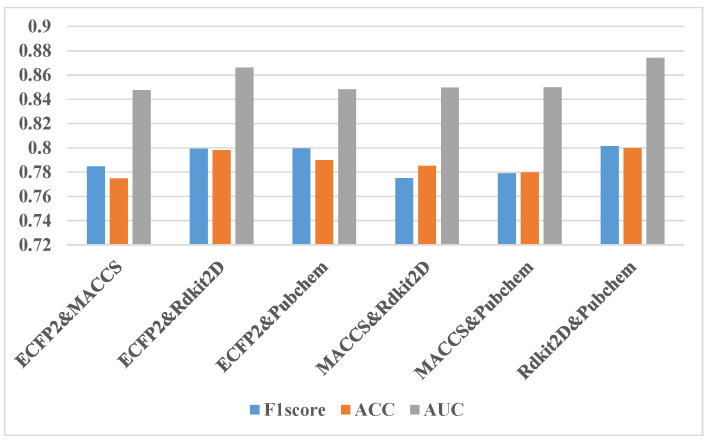
Results of ablation experiments by using only two fingerprints. “ECFP2&MACCS” means using the ECFP2 and MACCS part of the features from all the features, and so on for other labels.

**Figure 8 molecules-27-03112-f008:**
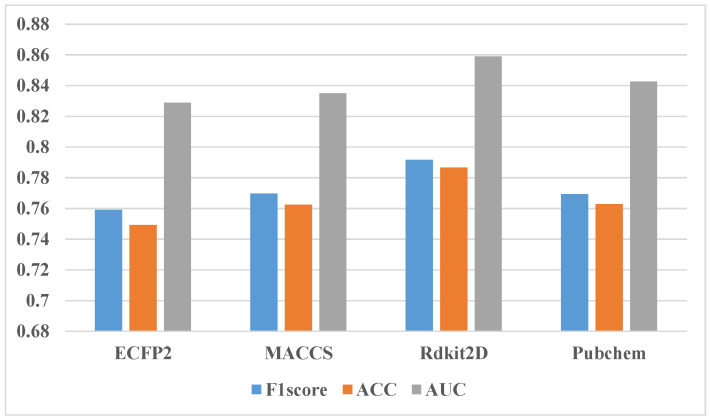
Results of ablation experiments by using only one fingerprint.

**Figure 9 molecules-27-03112-f009:**
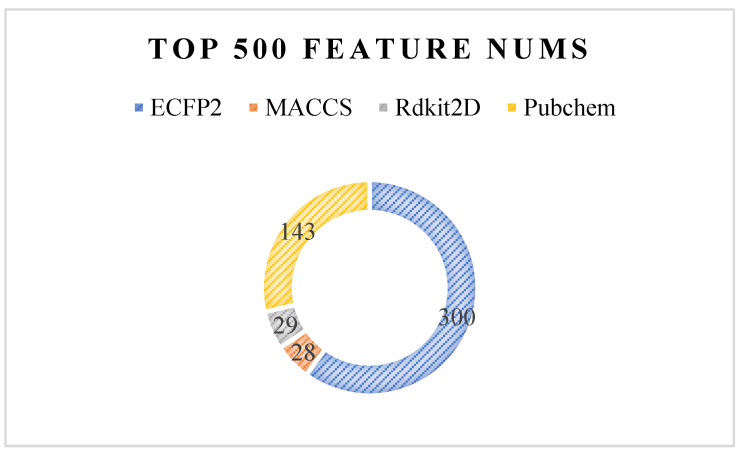
Results of the top 500 features of feature importance were obtained by counting the frequency of features in k feature subsets of all individuals in the last generation.

**Figure 10 molecules-27-03112-f010:**
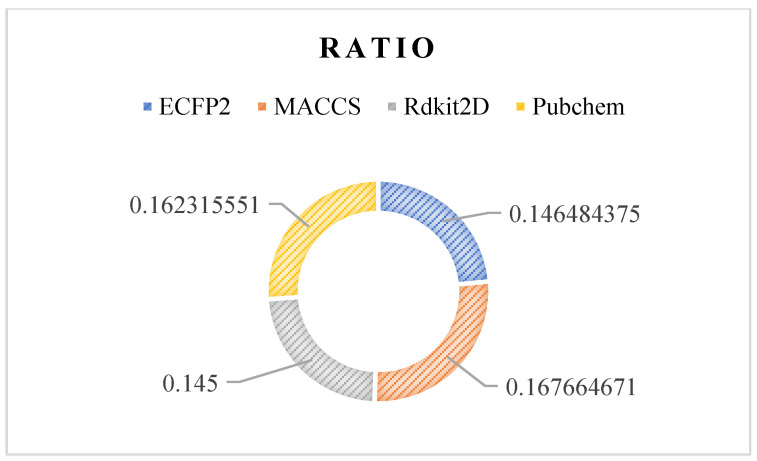
The ratios obtained by dividing the number of features of various fingerprints in the top 500 by the number of their original inputs are compared.

**Table 2 molecules-27-03112-t002:** Comparison of ACC, F1-score, and AUC results.

Models	ACC	F1-Score	AUC
SVC	0.747	0.685	0.766
Random Forest	0.760	0.700	0.782
XGBoost	0.723	0.669	0.740
AttentiveFP	0.729	0.716	0.750
GCN	0.725	0.698	0.759
GIN_AttrMasking	0.732	0.697	0.785
GIN_ContextPred	0.754	0.703	0.790
Voting Ensemble	0.753	0.752	0.826
**R-E-GA**	**0.770**	**0.769**	**0.842**

**Table 3 molecules-27-03112-t003:** Summary of datasets used in the external validation study.

	Datasets	DILI-Positive	DILI-Negative	Total Number
Training	Combined training dataset [6]	236	239	475
External validation	Combined validation dataset	114	84	198

**Table 4 molecules-27-03112-t004:** Performance of training using Mold2 and PaDEL descriptors in external validation study.

Molecular Descriptors	Index	Neural Network [6]		Xu et al. Model [6]		R-E-GA	
		10-Fold Test	Validation	10-Fold Test	Validation	10-Fold Test	Validation
**Mold2** **descriptors**	ACC	0.825	0.823	0.832	0.833	**0.852**	**0.851**
	AUC	-	0.916	-	0.931	0.912	**0.949**
	SEN	0.784	0.711	0.831	0.790	**0.855**	**0.794**
	SPE	**0.866**	0.976	0.833	0.893	0.850	**0.898**
PaDEL descriptors	ACC	0.816	0.791	0.823	0.811	**0.840**	**0.821**
	AUC	-	0.869	-	0.895	0.906	**0.904**
	SEN	0.758	0.723	0.852	**0.821**	**0.831**	0.797
	SPE	**0.875**	0.881	0.794	0.798	0.853	**0.858**

**Table 5 molecules-27-03112-t005:** Performance of external validation datasets training by Combined dataset with molecular One-hot Encoded representation.

Models	Internal 10-Fold Cross Validation				External Validation			
	ACC	AUC	SEN	SPE	ACC	AUC	SEN	SPE
**Xu et al. [6]**	**0.884**	**-**	**0.899**	**0.87**	**0.869**	**0.955**	**0.825**	**0.929**
R-E-GA	0.702	0.721	0.803	0.690	0.711	0.725	0.780	0.663
Number of drugs	(positive/negative = 236/239)				(positive/negative = 114/184)			

## Data Availability

The DILI data and source code for R-E-GA in this study are available at https://github.com/MLDMXM2017/R-E-GA, accessed on 1 April 2022.

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
