# Peer review of "An Algorithm Framework for Drug-Induced Liver Injury Prediction Based on Genetic Algorithm and Ensemble Learning"

_molecules, 2022, doi:10.3390/molecules27103112_

Round 1
Reviewer 1 Report
The authors propose an application of optimization and classification, with illustrative validation in the medical field. The paper answers positively the following general comments: Does the introduction provide sufficient background and include all relevant references? Is the research design appropriate? Are the methods adequately described? Are the results clearly presented? Are the conclusions supported by the results?. The following specific comments are formulated for its improvement:
1. Please edit the paper carefully such that to respect the instructions for authors. A homogeneous style is desired.
2. You should present the contributions with respect to your past papers that should be cited. Your past algorithms are very well appreciated.
3. Besides, the authors should include the following recent optimization algorithms as they proved to be successful in various applications, not necessarily medical ones: Gravitational search algorithm-based tuning of fuzzy control systems with a reduced parametric sensitivity (AICS 2011), HARD: Bit-split string matching using a heuristic algorithm to reduce memory demand (ROMJIST 2020), Policy iteration reinforcement learning-based control using a grey wolf optimizer algorithm (INS 2022).
4. I had a problem on understanding how the algorithm works exactly from the provided description.
5. You should also present the optimization problem that is solved by the genetic algorithm.
6. The connection of the optimization algorithm to the optimization problem is not pointed out currently with sufficient clarity.
7. Also, the connection classification problem - optimization problem requires better highlighting.
9. The connection of the application to the previous theory is not clear enough. More details are necessary for improved transparency.
10. You should save the code of programs and datasets, and cite the link to them in the paper. This is useful for validation, and helps the above comment. The importance of this comment is related to the fact that similar optimization algorithms are reported in the literature, they report excellent results but cannot be tested. The dataset is shared but the programs not. That will ensure a full transparency, but it is not a strict requirement.
11. You should specify which are the parameters of the optimization algorithm, which of them should be selected by the user and which of them are random.
12. The comparison with other optimization algorithms is also affected by the comments 10 and 11. Anyway, other values of parameters lead to different performance.
Author Response
Response to Reviewer 1 Comments
The authors propose an application of optimization and classification, with illustrative validation in the medical field. The paper answers positively the following general comments: Does the introduction provide sufficient background and include all relevant references? Is the research design appropriate? Are the methods adequately described? Are the results clearly presented? Are the conclusions supported by the results?. The following specific comments are formulated for its improvement:
Response : Thank you very much for your comments. We have read your comment carefully and revised the manuscript carefully.
Point 1:Please edit the paper carefully such that to respect the instructions for authors. A homogeneous style is desired.
Response 1: We carefully adjusted the Figures and Tables in the article based on the journal homogeneous style.
Point 2: You should present the contributions with respect to your past papers that should be cited. Your past algorithms are very well appreciated.
Response 2: Thank you for comments. At present, we have described and cited the past related papers of Genetic Algorithm and Rotation Forest.
Point 3: Besides, the authors should include the following recent optimization algorithms as they proved to be successful in various applications, not necessarily medical ones: Gravitational search algorithm-based tuning of fuzzy control systems with a reduced parametric sensitivity (AICS 2011), HARD: Bit-split string matching using a heuristic algorithm to reduce memory demand (ROMJIST 2020), Policy iteration reinforcement learning-based control using a grey wolf optimizer algorithm (INS 2022).
Response 3: Thank you very much for your recommendation. We have read these three articles carefully and quoted them in the articles. Based on these three papers, it can be concluded that the search optimization algorithm has excellent applications in many fields. It has the characteristics of global search caused by low parameter sensitivity and randomness. In view of the fact that the introduction part of our article does not introduce much about the optimization algorithm, we have cited these three papers to supplement the Introduction chapter on the application of optimization algorithms in various fields.
Point 4: I had a problem on understanding how the algorithm works exactly from the provided description.
Response 4: The overall idea of R-E-GA integrates Rotation, Ensemble Learning and Genetic Algorithm. The role of Rotation is to rotate the feature space and map the data to a more distinguishable space. The role of Ensemble Learning is to jointly vote with k weak classifiers to increase the fault tolerance and correction ability of the model. Genetic Algorithms are used to search and optimize feature subsets suitable for Rotation and Ensemble operations. It is only part of R-E-GA.
How the three algorithm operations are integrated is described as follows.
Genetic Algorithm optimizes the solution by simulating natural selection through crossover operator, mutation operator and elimination selection operator. The solution here is the individual in GA. Each individual represents an ensemble classifier trained on a subset of features. The ensemble classifier mentioned here is an Adaboost-type ensemble classifier, which means that each weak classifier in the ensemble classifier adjusts the sample weight on the performance of the previous classifier. So each individual is first a feature subset. K weak classifiers perform data space rotation mapping and model training on this feature subset. And finally the results of the k weak classifiers are ensembled as the final training ensemble classifier for this individual. So the content that GA searches and optimizes is the feature subset.
Next, the algorithm flow is briefly described. The over flow of R-E-GA shown in the figure below is carried out according to GA framework.
M feature subsets are randomly generated
After rotating the feature space formed by the feature subsets, the ensemble classifier training of k weak classifiers is serially integrated.
Individual evaluation is performed on M individuals, and the evaluation standard is the classification performance of the ensemble classifier composed of individuals.
Cross and mutate the M feature subsets to randomly search for solutions in the solution space.
Loop until GA ends.
In this way, as long as the number of individuals and evolutionary algebra are sufficient, the solution searched when algorithm ends is the approximate global optimal solution.
Point 5: You should also present the optimization problem that is solved by the genetic algorithm.
Response 5: In the actual algorithm operation, the object of GA search optimization is the feature subset, so the problem solved by the optimization algorithm is to search for the feature subset with better performance. However, since R-E-GA designs the individual optimized by GA as an ensemble classifier. K weak classifiers are trained after rotation based on the current feature subset in the ensemble classifier. So the optimization algorithm finally searches for a better performing ensemble classifier by searching for a better feature subset. We add the following to the subsection 2.4 Framework of Rotation-Ensemble-GA Algorithm in the manuscript to explain the optimization problem solved by R-E-GA.
“The search finds a better feature subset through the continuous evolution of the combina-tion and rotation of feature subsets, so that the ensemble classifier is more effective. The solution is continuously optimized in this way. When the evolution ends, the optimal in-dividual is regarded as the approximate global optimal solution.”
Point 6: The connection of the optimization algorithm to the optimization problem is not pointed out currently with sufficient clarity
Response 6: This study defines the optimization problem as an ensemble classifier with sufficiently effective classification performance for DILI data. This ensemble classifier can be obtained through GA search optimization. The unit of R-E-GA search optimization is feature subsets. Therefore, it can be considered that R-E-GA optimizes the ensemble classifier by searching for a subset of features suitable for rotation and ensemble, and further optimizes the DILI prediction performance. We add the following to the subsection 2.4 Framework of Rotation-Ensemble-GA Algorithm in the manuscript to explain the optimization problem solved by R-E-GA.
“In R-E-GA, each ensemble classifier containing K weak classifiers is used as an individual for evolution. Find new solutions through crossover and mutation operations. Select and retain the individuals with better performance by F1-score. In this way, the solution in the candidate set is in a process of gradual optimization.”
Point 7: Also, the connection classification problem - optimization problem requires better highlighting.
Response 7: The connection between classification problem and search optimization problem has been highlighted in the article.
(I'm sorry we didn't notice Point 8 in the Comment)
Point 9: You should also present the optimization problem that is solved by the genetic algorithm.
Response 9: The Introduction chapter has been supplemented with optimization problems in numerous domains solved by genetic algorithms, such as micro-expression recognition and search optimization algorithm improvements combined with ECOC. The relevant papers have been supplemented and cited. This study purpose to use genetic algorithm to optimize the search of feature subsets and ensemble classifiers. The specific optimization process has been presented in the evolution curve.
Point 10: You should save the code of programs and datasets, and cite the link to them in the paper. This is useful for validation, and helps the above comment. The importance of this comment is related to the fact that similar optimization algorithms are reported in the literature, they report excellent results but cannot be tested. The dataset is shared but the programs not. That will ensure a full transparency, but it is not a strict requirement.
Response 10: Thank you very much for your suggestion. We have further described the data in part of Datasets, and the DILI data and source code for R-E-GA in this study is available at https://github.com/MLDMXM2017/R-E-GA. We hope that our experimental results and algorithm can be verified and helpful for the DILI prediction.
Point 11: You should specify which are the parameters of the optimization algorithm, which of them should be selected by the user and which of them are random.
Response 11: Parameter descriptions have been supplemented in the article. As an overall algorithm framework, R-E-GA follows the process of the GA but does not isolate the optimization algorithm alone. The optimization algorithm is only part of R-E-GA. Therefore, the parameters proposed in this paper are all user-selectable parameters. Strictly speaking, Iteration size, Population size and Possibility of Mutation are fixed parameters in the optimization algorithm GA process, and the other parameters are added for R-E-GA. However, they can all be set by the user.
Point 12: The comparison with other optimization algorithms is also affected by the comments 10 and 11. Anyway, other values of parameters lead to different performance.
Response 12: R-E-GA and the optimization algorithm used in this study are indeed affected by parameter settings, but R-E-GA is a random search algorithm in the solution space. So the influence of parameters is significantly smaller than that of the comparison algorithm. In the comparison experiment, we also improved the performance of the comparison optimization algorithm experiment through several parameter adjustment experiments. The focus of R-E-GA is not only optimization, which is only part of algorithm. And the application of ensemble is also the reason for a large part of the performance improvement. Thank you very much for your comment. Due to author authorization, the benchmark comparison algorithm used in this paper comes from Sci-Kit Learning Python package[1], and the algorithm code based on graph neural network comes from literature [2, 3], which you can obtain from available access: https://github.com/kexinhuang12345/DeepPurpose.
Reference
- Pedregosa, F.; Varoquaux, G.; Gramfort, A.; Michel, V.; Thirion, B.; Grisel, O.; Blondel, M.; Prettenhofer, P.; Weiss, R.; Dubourg, V.; Vanderplas, J.; Passos, A.; Cournapeau, D.; Brucher, M.; Perrot, M.; Duchesnay, E., Scikit-learn: Machine Learning in Python. J Mach Learn Res 2011, 12, 2825-2830.
- Huang, K.; Fu, T.; Glass, L. M.; Zitnik, M.; Xiao, C.; Sun, J., DeepPurpose: a deep learning library for drug-target interaction prediction. Bioinformatics 2021, 36, (22-23), 5545-5547.
- Li, M. F.; Zhou, J. J.; Hu, J. J.; Fan, W. X.; Zhang, Y. K.; Gu, Y. X.; Karypis, G., DGL-LifeSci: An Open-Source Toolkit for Deep Learning on Graphs in Life Science. Acs Omega 2021, 6, (41), 27233-27238.

Reviewer 2 Report
This is an article to address An Algorithm Framework for Drug-induced Liver Injury Prediction based on Genetic Algorithm and Ensemble Learning. It was an interesting new algorithm.
The font in the Figure is different, please correct it.
Author Response
Response to Reviewer 2 Comments
Point 1: The font in the Figure is different, please correct it.
Response 1: We thank the reviewer’s positive comment. We carefully revised and unified the font in all Figures. Also modified Figures 6-10 for better presentation. And we will pay attention to such problems in future studies. Previously, Figure 6-8 had the problem that the font did not match the font of other figures, and the two curves were too overlapping. At present, we have modified Figure 6-8 into a bar chart and unified the font format.

Reviewer 3 Report
Drug-induced liver injury (DILI) is among the most unpredictable adverse reactions to xenobiotics in humans and the most frequent cause of approved drugs withdrawals. The authors have proposed a sophisticated genetic algorithm to tackle this problem, and rigorously examined it against a rich data source, the results appear to be promising and may bring new insights for other research groups working on DILI studies. The manuscript is written with a high standard, the algorithm descriptions are comprehensive and clear.
The reviewer believes the manuscript overall meets a high standard, albeit a few very minor issues if the authors get the chance to address it:
- The data source collected for this study is comprehensive and carefully preprocessed. Perhaps there is a need to elaborate and reassure in more details on the binarization logic since some of the original dataset had multiple categories.
- It would also be interesting to see how the performance of the proposed algorithm compares against the results of those papers (e.g., ref 6-8), by applying to their original data source.
- Supplementary data set is provided as non-published material, since the data source are public domain, would it be possible to make it available to public so that peers may further benefit from this work.
- L100: the csv file provided contains 1498 positive & 1433 negative.
- L295: has a typo.
- L429: gramma issue.
- Eq. 8 has formatting issue.
- Figure 6-8: the blue curves in these 3 figures are awkwardly coincided with the orange curve, making them difficult to be noticed. Perhaps there is a better way to visualize them.
Author Response
Response to Reviewer 3 Comments
Drug-induced liver injury (DILI) is among the most unpredictable adverse reactions to xenobiotics in humans and the most frequent cause of approved drugs withdrawals. The authors have proposed a sophisticated genetic algorithm to tackle this problem, and rigorously examined it against a rich data source, the results appear to be promising and may bring new insights for other research groups working on DILI studies. The manuscript is written with a high standard, the algorithm descriptions are comprehensive and clear.
The reviewer believes the manuscript overall meets a high standard, albeit a few very minor issues if the authors get the chance to address it:
Response : Thank you so much for your positive comments. And the comments you mentioned have been very helpful to us.
Point 1: The data source collected for this study is comprehensive and carefully preprocessed. Perhaps there is a need to elaborate and reassure in more details on the binarization logic since some of the original dataset had multiple categories.
Response 1: According to your comment, we elaborated and reassure in more detail on the binarization logic at part of Dataset. This is of great help to the access and credibility of DILI data set. The specific revised are as follows:
“Binarize the labels of different datasets to obtain binary labels. The rules of the label are shown in Table 1. We adopt cautious binarization rules and took compounds with high reliability DILI classes. First, the data comes from trusted sources such as scientific literature, medical monographs clinical and data approved by FDA. Second, we set labels to “1” for the compound with definite DILI from the original source, and “0” for the compound without DILI from the original source. This is reflected in the data processing of Greene, DILIrank, Livertox and LTKB data set, where “HH” and “Most concern” represent “Evidence of human hepatotoxicity” and DILI-positive respectively. Meanwhile, “NE” and “no concern” indicate “no evidence of hepatotoxicity in any species” and DILI-negative [42-47]. The “Category A”and “Category B” from the LiverTox are the classes of compounds that have been “frequently reported” and "reported" to cause DILI, "Category E" means “no evidence that the drug has caused liver injury” [44, 48]. We find that Shuaibing et al. and Mulliner et al. had the same strict binarization rules as we adopted , so we considered the data of these authors also to be credible [7, 8]. It is found that Xu et al. 's binding data also came from highly trusted data sources, including NCTR, Greene et al, and Xu et al., which removed inconsistent compound’s DILI label from the dataset, and we considered their data equally reliable. [6] Finally, to expand the data set, we take a small portion compounds from Greene's "WE" compound’s DILI classes which represents “Weak evidence of (<10 case reports) of human hepatotoxicity”, and they are also considered as DILI compounds in the literature.[44, 47]”
Table 1. Datasets of DILI and binarization rules of labels.
|
ID |
Source |
Type of Data |
No. of Compound (Positive/Negative) |
DILI Categories |
|
1 |
Greene et al.,2010[1] |
Literature reviews and medical monographs |
487(331/156) |
HH, WE represented positives and NE represented negatives |
|
2 |
Xu et al.,2015[2] |
Medical monographs and FDA-approved drug labeling |
475(236/239) |
Authors definition |
|
3 |
Mulliner et al,2017[3] |
Clinical data and drug labeling |
1370(932/438) |
Authors definition |
|
4 |
Shuaibing et al,2019[4] |
Drug labeling and comprehensive data |
1458(761/697) |
Authors definition |
|
5 |
DILIrank[5] |
Drug labeling and clinical data |
504(192/312) |
Most concern as 1; no concern as 0 |
|
6 |
Livertox[6] |
Scientific literature and public database |
343(119/224) |
Categories A and B were combined into positives, and Category E was considered as negatives |
|
7 |
LTKB[7] |
FDA-approved drug labeling |
195(113/82) |
Most concern as 1;no concern as 0 |
Point 2: It would also be interesting to see how the performance of the proposed algorithm compares against the results of those papers (e.g., ref 6-8), by applying to their original data source.
Response 2: Thank you so much for this comment. We collected the labels and computational features of the original data source. This response is divided into two parts A and B. In Part A, data acquisition is discussed, and in part B, experimental results are expressed and explained.
- Original DILI Data (label and feature) available access situation from ref.6-8[2-4]
The data sets used in reference 6-8 and their computational feature are shown in Table 2.
Table2. Original Data set and publicly available situation according to Ref. 6-8
|
Ref. |
Author |
Data sets |
Feature |
Public available |
|
6 |
Xu et al.[2] |
Training data set (3): NCTR, Liew, Combined External validation (5): NCTR validation, Liew validation, Combined validation, Greene, Xu validation |
One-hot Encoded, Mold2 descriptors, PaDEL descriptors |
Yes |
|
7 |
Mulliner et al.[3] |
Author definition |
CATS, MOE, MDL. VolSurf+ (descriptor) |
MOE and VolSurf+ is not public available |
|
8 |
He et al.[4] |
Author definition |
Jchem phychem properties |
Jchem is not available |
Due to copyright and public data available issues, we were only able to get all the raw data from Ref 6 and experiment without using the original data from Ref. 7-8.[3, 4] The sources of unavailable computing features come from references 7-8, and paid software and unavailable data were shown below.
Volsurf+ official link: https://www.sciencedirect.com/science/article/pii/S0166128099003607[3]
MOE official link: https://www.chemcomp.com/Products.htm
Jchem[4]: Jchem has two versions, free and commercial, but the free version requires a certificate, and our certificate application has not been approved for the time being, as shown in follow figure.
Therefore, we compared the original data set of reference 6.
- Comparison of R-E-GA and Xu et al.’s model on the data set from rer.6, i.e., Xu et al.[2]
In this section, three computational features, i.e., One-hot encoded[2, 8], Mold2[9] descriptor, PaDEL[10] descriptor were used for comparison on the Combined data set as reference 6. The Combined data set compiled by 4 original data sets sources, that is, NCTR[11, 12], Liew[13], Greene[1], Xu[14]. Xu et al. used the original UGRNN[8] and DNN algorithm for model building on the Combined data set[2]. And the label of Ref.6 we compared was shown as Table3. Xu's main experimental results and three computational features were compared on this data set, and we performed the calculations under exactly the same Settings. As results, our model R-E-GA’s performance is superior to Xu’s on the data set with Mold2 , PaDEL molecular descriptors and inferior to Xu’s model on the data set of One-hot encoding by SMILES, as shown in Table 3-5. In addition, on the same data set, R-E-GA based on single descriptor Mold2 and deep learning based on one-hot encoded model were basically consistent with index AUC (0.949 to 0.955).
Table 3. Summary of data sets used in the external validation study
|
|
Data sets |
DILI-positive |
DILI-negative |
Total number |
|
Training |
Combined training data set[2] |
236 |
239 |
475 |
|
External validation |
Combined validation data set |
114 |
84 |
198 |
We first compared the computational models of Xu et al. based on molecular descriptors on Combined data set. In original article, Xu et al. calculates Mold2(777) [9] and PaDEL(1444) [10] two molecular descriptors for model trainning on Combined data set respectively. We also calculated Mold2 and PaDEL descriptors as model inputs and the results to comparise algorithms were shown in Table 4. And we use the same 10-fold cross validation and evaluation index as ACC, AUC, SEN, SPE for modeling in above studies. The prediction results of R-E-GA on Combined data set with Mold2 descriptor were an ACC of 0.852, SEN of 0.855, SPE of 0.850 on 10-fold test data set, and obtained the performance ACC of 0.851, AUC of 0.949, SEN of 0.794, SPE of 0.898 on the external data set, which are about average 0.02 index higher than Xu’s model. This is consistent with our other experimental results. The R-E-GA model based on molecular fingerprint(descriptor) has excellent DILI prediction performance.
Table 4. Performance of external validation datasets training by Combined data set with Mold2 and PaDEL descriptors
|
Molecular descriptors |
index |
Neural network |
Xu et al. model |
R-E-GA |
|||
|
10-fold test |
validation |
10-fold test |
validation |
10-fold test |
validation |
||
|
Mold2 descriptors |
ACC |
0.825 |
0.823 |
0.832 |
0.833 |
0.852 |
0.851 |
|
AUC |
- |
0.916 |
- |
0.931 |
0.912 |
0.949 |
|
|
SEN |
0.784 |
0.711 |
0.831 |
0.790 |
0.855 |
0.794 |
|
|
SPE |
0.866 |
0.976 |
0.833 |
0.893 |
0.850 |
0.898 |
|
|
PaDEL descriptors |
ACC |
0.816 |
0.791 |
0.823 |
0.811 |
0.840 |
0.821 |
|
AUC |
- |
0.869 |
- |
0.895 |
0.906 |
0.904 |
|
|
SEN |
0.758 |
0.723 |
0.852 |
0.821 |
0.831 |
0.797 |
|
|
SPE |
0.875 |
0.881 |
0.794 |
0.798 |
0.853 |
0.858 |
|
At the same time, we compared the model calculation results of molecules one-hot encoding to Xu et al.[2] In Xu et al. 's model, atom types are encoded as C = (1,0,0), N = (0,1,0), O = (0,0,1) , and bond types are similarly encoded[2, 8]. The same molecular encoding method was adopted in our model, and the performance of comparison to R-E-GA on Combined data set was shown in Table 5. We find that the accuracy of the model is inferior to that of the Xu’s model when the one-hot vector encoded by SMILES is used as the calculation feature for DILI prediction. We thought the One-hot Encoder methods can be regarded as the representation extracted from the one dimensional linear representation of molecules’s SMILES[2, 15]. The model R-E-GA cannot extract toxicity fingerprints such as physicochemical properties and pharmacophore from linear representation based on SMILES for DILI prediction.
Table 5. Performance of external validation datasets training by Combined data set with molecular One-hot Encoded representation
|
Models |
Internal 10-fold cross validation |
External validation |
||||||
|
ACC |
AUC |
SEN |
SPE |
ACC |
AUC |
SEN |
SPE |
|
|
Xu et al.[2] |
0.884 |
- |
0.899 |
0.87 |
0.869 |
0.955 |
0.825 |
0.929 |
|
R-E-GA |
0.702 |
0.721 |
0.803 |
0.690 |
0.711 |
0.725 |
0.780 |
0.663 |
|
Number of drugs |
(positive/negative = 236/239) |
(positive/negative = 114/184) |
||||||
As the results, we can conclude that R-E-GA performs best in descriptors of Modl2 and PaDEL. Each metrics can achieve the best performance in each dataset. But in the One-hot Encoder vector for molecules, the performance is more moderate, the results of R-E-GA is worse than Xu et al.’s. The result was in line with our expectations. R-E-GA is designed as a DILI prediction model for the multiple molecular fingerprint fusion method, and it has obvious computational advantages for high-dimensional fusion representations data.So, R-E-GA have well-performed results on Mold2 and PaDEL descriptors compare with Xu et al. ‘s deep learning model, as shown in Table 4 [6] [2]. They are enough to verify R-E-GA’s ability in feature extraction and model performance of DILI prediction. The above results show that R-E-GA has obvious advantages in the processing of DILI prediction model based on molecular descriptors, but it has certain limitations in the processing of linear representation, and we already proved the complexity of multiple molecular representations fusion is conducive to reflect the superiority of R-E-GA.
Point 3: Supplementary data set is provided as non-published material, since the data source are public domain, would it be possible to make it available to public so that peers may further benefit from this work.
Response 3: Thanks for your suggestion. We have uploaded the DILI data in our study at https://github.com/MLDMXM2017/R-E-GA. The data is available to the public and the link have been updated to the manuscript.
Point 4-7 Minor issues:
- L100: the csv file provided contains 1498 positive & 1433 negative.
- L295: has a typo.
- L429: gramma issue.
- 8 has formatting issue.
Response: Thanks you for reading so carefully. We have revised the corresponding parts in the manuscript.
Point 8: Figure 6-8: the blue curves in these 3 figures are awkwardly coincided with the orange curve, making them difficult to be noticed. Perhaps there is a better way to visualize them.
Response 8: We decided to change the line graph in Figure 6-8 to bar graph in the manuscript. It would not affect the comparison of the results of the ablation experiments, but could also visualize the differences between the various metrics clearly.
(Finally, we also thank the academic editor for his/her positive comments and support.)
Reference1. Greene, N.; Fisk, L.; Naven, R. T.; Note, R. R.; Patel, M. L.; Pelletier, D. J., Developing Structure-Activity Relationships for the Prediction of Hepatotoxicity. Chemical Research in Toxicology 2010, 23, (7), 1215-1222.
- Xu, Y.; Dai, Z.; Chen, F.; Gao, S.; Pei, J.; Lai, L., Deep Learning for Drug-Induced Liver Injury. J Chem Inf Model 2015, 55, (10), 2085-93.
- Mulliner, D.; Schmidt, F.; Stolte, M.; Spirkl, H. P.; Czich, A.; Amberg, A., Computational Models for Human and Animal Hepatotoxicity with a Global Application Scope. Chem Res Toxicol 2016, 29, (5), 757-67.
- He, S.; Ye, T.; Wang, R.; Zhang, C.; Zhang, X.; Sun, G.; Sun, X., An In Silico Model for Predicting Drug-Induced Hepatotoxicity. Int J Mol Sci 2019, 20, (8).
- Chen, M. J.; Suzuki, A.; Thakkar, S.; Yu, K.; Hu, C. C.; Tong, W. D., DILIrank: the largest reference drug list ranked by the risk for developing drug-induced liver injury in humans. Drug Discov Today 2016, 21, (4), 648-653.
- Hoofnagle, J. H.; Serrano, J.; Knoben, J. E.; Navarro, V. J., LiverTox: A website on drug-induced liver injury. Hepatology 2013, 57, (3), 873-874.
- Chen, M.; Zhang, J.; Wang, Y.; Liu, Z.; Kelly, R.; Zhou, G.; Fang, H.; Borlak, J.; Tong, W., The Liver Toxicity Knowledge Base: A Systems Approach to a Complex End Point. Clin Pharmacol Ther 2013, 93, (5), 409-412.
- Lusci, A.; Pollastri, G.; Baldi, P., Deep Architectures and Deep Learning in Chemoinformatics: The Prediction of Aqueous Solubility for Drug-Like Molecules. Journal of Chemical Information and Modeling 2013, 53, (7), 1563-1575.
- Hong, H. X.; Xie, Q.; Ge, W. G.; Qian, F.; Fang, H.; Shi, L. M.; Su, Z. Q.; Perkins, R.; Tong, W. D., Mold(2), molecular descriptors from 2D structures for chemoinformatics and toxicoinformatics. Journal of Chemical Information and Modeling 2008, 48, (7), 1337-1344.
- Yap, C. W., PaDEL-Descriptor: An Open Source Software to Calculate Molecular Descriptors and Fingerprints. J Comput Chem 2011, 32, (7), 1466-1474.
- Chen, M. J.; Hong, H. X.; Fang, H.; Kelly, R.; Zhou, G. X.; Borlak, J.; Tong, W. D., Quantitative Structure-Activity Relationship Models for Predicting Drug-Induced Liver Injury Based on FDA-Approved Drug Labeling Annotation and Using a Large Collection of Drugs. Toxicol Sci 2013, 136, (1), 242-249.
- Chen, M. J.; Vijay, V.; Shi, Q.; Liu, Z. C.; Fang, H.; Tong, W. D., FDA-approved drug labeling for the study of drug-induced liver injury. Drug Discov Today 2011, 16, (15-16), 697-703.
- Liew, C. Y.; Lim, Y. C.; Yap, C. W., Mixed learning algorithms and features ensemble in hepatotoxicity prediction. J Comput Aid Mol Des 2011, 25, (9), 855-871.
- Xu, J. H. J.; Henstock, P. V.; Dunn, M. C.; Smith, A. R.; Chabot, J. R.; de Graaf, D., Cellular imaging predictions of clinical drug-induced liver injury. Toxicol Sci 2008, 105, (1), 97-105.
- Atz, K.; Grisoni, F.; Schneider, G., Geometric deep learning on molecular representations. Nat Mach Intell 2021, 3, (12), 1023-1032.

Round 2
Reviewer 1 Report
The paper is improved. It deserves to be published although the github link does not work.